# Diagnostic Performance of Immunohistochemistry Compared to Molecular Techniques for Microsatellite Instability and p53 Mutation Detection in Endometrial Cancer

**DOI:** 10.3390/ijms24054866

**Published:** 2023-03-02

**Authors:** Sylvie Streel, Alixe Salmon, Adriane Dheur, Vincent Bours, Natacha Leroi, Lionel Habran, Katty Delbecque, Frédéric Goffin, Clémence Pleyers, Athanasios Kakkos, Elodie Gonne, Laurence Seidel, Frédéric Kridelka, Christine Gennigens

**Affiliations:** 1Department of Medical Oncology, CHU Liège, 4000 Liège, Belgium; 2Department of Gynecology and Obstetrics, CHU Liège, 4000 Liège, Belgium; 3Department of Human Genetics, CHU Liège, 4000 Liège, Belgium; 4Department of Pathology, CHU Liège, 4000 Liège, Belgium; 5Department of Radiotherapy Oncology, CHU Liège, 4000 Liège, Belgium; 6Department of Biostatistics, CHU Liège, 4000 Liège, Belgium

**Keywords:** endometrial cancer, molecular classification, immunohistochemistry, polymerase chain reaction, next generation sequencing method, p53, *TP53*, mismatch repair, microsatellite instability

## Abstract

Molecular algorithms may estimate the risk of recurrence and death for patients with endometrial cancer (EC) and may impact treatment decisions. To detect microsatellite instabilities (MSI) and p53 mutations, immunohistochemistry (IHC) and molecular techniques are used. To select the most appropriate method, and to have an accurate interpretation of their results, knowledge of the performance characteristics of these respective methods is essential. The objective of this study was to assess the diagnostic performance of IHC versus molecular techniques (gold standard). One hundred and thirty-two unselected EC patients were enrolled in this study. Agreement between the two diagnostic methods was assessed using Cohen’s kappa coefficient. Sensitivity, specificity, positive (PPV) and negative predictive values (NPV) of the IHC were calculated. For MSI status, the sensitivity, specificity, PPV and NPV were 89.3%, 87.3%, 78.1% and 94.1%, respectively. Cohen’s kappa coefficient was 0.74. For p53 status, the sensitivity, specificity, PPV, and NPV were 92.3%, 77.1%, 60.0% and 96.4%, respectively. Cohen’s kappa coefficient was 0.59. For MSI status, IHC presented a substantial agreement with the polymerase chain reaction (PCR) approach. For the p53 status, the moderate agreement observed between IHC and next generation sequencing (NGS) methods implies that they cannot be used interchangeably.

## 1. Introduction

Endometrial cancer (EC) is the sixth most frequent female cancer, affecting mainly post-menopausal women [1]. The incidence of EC increased by 132% between 1999 and 2019, with the highest progression in developed countries. In contrast to EC incidence, EC mortality rates significantly decreased in around half of countries, and the mortality-to-incidence ratio decreased worldwide. This is the result of a better understanding of EC pathology and more effective treatments [2].

Recent integration of molecular analysis provides insights into disease biology and improves the diagnosis and risk stratification of patients with EC. This new approach allows clinicians to individualise therapeutic management, particularly for adjuvant treatments, but also in the metastatic setting [3,4].

The stepwise diagnostic algorithm, approved by the World Health Organization (WHO) in 2020, categorises EC into 4 molecular subgroups [5]. The first step of this diagnostic algorithm is to identify EC with a pathogenic mutation in polymerase-E exonuclease domain. This leads to the first “*POLE* ultra-mutated” (*POLE*mut) subgroup. This subgroup has an excellent prognosis [6]. The second assessment amongst *POLE* wildtype (*POLE*wt) tumours is the loss of expression in one or more of the 4 mismatch repair (MMR) proteins (MLH1, MSH2, MSH6 or PMS2) categorised as the MMR-deficient (MMRd) EC. This second subgroup, the MMRd, has an intermediate prognosis with good response to immune checkpoints inhibitors when the EC is at an advanced or recurrent stage (for early stages no conclusion has yet been published) [7]. Finally, the abnormal expression of p53 is investigated on MMR-proficient (MMRp) tumours to determine EC with (p53-abnormal–3rd subgroup) or without (nonspecific molecular profile/NSMP–4th subgroup) p53 anomalies [8,9]. The p53-abnormal subgroup is the most aggressive and lethal molecular subtype. Recent data suggest that patients with p53-abnormal EC benefit from adjuvant treatment intensification with chemoradiotherapy followed by adjuvant chemotherapy [10].

Concerning analysis methods, the identification of *POLE*mut is exclusively performed using DNA sequencing methods [11]. The MMRd can be identified by immunohistochemistry (IHC), by molecular methods such as polymerase chain reaction (PCR) or by next generation sequencing (NGS) method. The PCR method consists of identifying the molecular hallmark of MMRd: microsatellite instability (MSI) [12]. To determine the p53 status, IHC or *TP53* NGS methods can be used [9].

In the case of the detection of MMR and p53 mutations, the IHC method could be preferred over molecular techniques because it is rapid, widely available, less expensive, requires less tumour material and is readily interpretable [9,11,12,13].

Discrepancies between the results obtained by each method have been described in several studies [9,12,13,14,15,16,17]. Given the importance of molecular classification at the time of diagnosis and its influence on all aspects of EC care (surgical decision and adjuvant/metastatic therapies) [18] and research, further studies are needed to investigate the performance characteristics of the methods used in this molecular classification. The results provide clinicians and researchers with some evidence to thereby choose the most appropriate method depending on their goals and resources.

The objective of the study was to assess the diagnostic performance of IHC in comparison with molecular analyses, considered as the gold standard, to determine MMR/MSI and p53 gene status.

## 2. Results

### 2.1. Patients and Tumour Characteristics

Based on a retrospective medical chart review, 166 patients were enrolled in the study; however, 34 patients were excluded due to missing data. The remaining 132 patients constituted the study population.

The clinicopathological characteristics of the 132 EC patients are summarised in Table 1. The median age was 69.0 years. The majority of the EC tumours had an endometrioid histology (82.3%), International Federation of Gynecology and Obstetrics (FIGO) tumour grade of 1 and 2 (65.9%), nodal stage N0 (91.5%) and FIGO stage IB (44.6%).

### 2.2. Molecular Profile of EC Tumours

The molecular profile of the 132 EC patients is reported in Table 2. Eleven patients (9.7%) had a *POLE*mut tumour.

MMR IHC analysis was performed on 131 EC tumours and revealed a MMRd status in 44 cases (33.6%). Among the 131 MMR IHC assays, 99 MSI PCR were carried out and 34 patients presented an MSI-high status (34.3%). Agreement between the two methods was thus evaluated on 99 subjects.

Concerning the determination of p53 status, p53 IHC testing was achieved in 129 EC tumours and showed 34 cases with an abnormal p53 status (26.4%). *TP53* sequencing was carried out on 106 EC tumours and identified 25 mutated cases (23.6%). Among the 129 p53 IHC tests, 104 cases were also examined by the *TP53* sequencing method. These cases were used to determine any agreement between the IHC and molecular analyses.

### 2.3. Agreement between MMR IHC Status and MSI PCR Testing

With the PCR analysis, MSI-high was observed in 34 of the 99 cases. When considering the MMR IHC analysis, 39 cases were classified as MMRd among 99 cases. Agreement between MSI PCR and MMR IHC analyses was observed in 88 of 99 cases using a binary classification (Table 3). The proportion of MMRd/MSI-high was not statistically different using both analysis methods (34.3% vs. 39.4%, *p* = 0.13). Cohen’s kappa coefficient was 0.76 (95% CI: 0.63–0.89).

Table 3 displays the performance of the MMR IHC method. Sensitivity was 91.2% (95% CI: 76.3–98.1) and specificity was 87.7% (95% CI: 77.2–94.5), yielding a global accuracy level of 88.9% (95% CI: 81.0–94.3). The positive predictive value (PPV) was 79.5% (95% CI: 63.5–90.7) while the negative predictive value (NPV) was 95.0% (95% CI: 86.1–99.0).

When analyses were performed on *POLE*wt tumours alone, as per the WHO algorithm, the accuracy of the MMR IHC method was 88.0% (95% CI: 79.0–94.1) with a sensitivity of 89.3% (95% CI: 71.8–97.7) and a specificity of 87.3% (95% CI: 75.5–94.7). Cohen’s kappa coefficient was 0.74 (95% CI: 0.59–0.89).

The profile for MLH1, PMS2, MSH6, MSH2, of the ten discordant cases is presented in Table 4. Among the seven cases considered as MSS using the MSI PCR method, one case lost expression of MSH6 protein and six cases lost expression of MLH1 and PMS2 according to the MMR IHC method. For the three remaining cases, although all proteins were present, MSI-high was detected by PCR.

### 2.4. Agreement between p53 IHC and TP53 NGS

With the sequencing analysis, *TP53* mutation was observed in 24 of the 104 cases. When considering the p53 IHC analysis, 30 out of 104 patients were classified as having an abnormal status. Therefore, the agreement between *TP53* NGS and p53 IHC analysis was observed in 86 of 104 cases using a binary classification (Table 5). The proportion of tumours with an abnormal status was the same between the two methods (23.1% vs. 28.9%, *p* = 0.16). Cohen’s kappa coefficient was 0.55 (95% CI: 0.37–0.73).

Table 5 displays the performance of the p53 IHC analysis. Sensitivity was 75.0% (95% CI: 53.3–92.0) and specificity was 85.0% (95% CI: 75.3–92.0), yielding a global accuracy level of 82.7% (95% CI: 74.0–89.4). The PPV was 60.0% (95% CI: 41.6–77.3) and the NPV was 92.0% (95% CI: 83.2–97.0).

When analyses were performed on *POLE*wt and MMRp cases (n = 48), the sensitivity increased to 92.3% (95% CI: 64.0–99.8) and the specificity decreased to 77.1% (95% CI: 59.9–89.6). The proportion of tumours with an abnormal status was lower with *TP53* analysis than with p53 IHC. This proportion was different between the two analysis methods (27.1% vs. 41.7%, *p* = 0.020). Cohen’s kappa coefficient was 0.59 (95% CI: 0.37–0.82).

The p53 IHC pattern of the 48 cases was matched with the type of *TP53* mutation on Table 6. The most prevalent abnormal p53 pattern was nuclear overexpression (14 out of 20–70%) of which 10 cases were found with a missense mutation in *TP53* and 4 cases with no *TP53* mutation. Complete absence of p53 expression was observed in 5 out of 20 cases. Among them, one case presented a stop gain mutation, the other cases had no *TP53* mutation. A subclonal pattern was observed in one tumour and was associated with a stop gain mutation. The false-negative case presented a missense mutation.

## 3. Discussion

This retrospective study included a cohort of unselected EC patients to assess the diagnostic performance of IHC compared with the molecular technique for the determination of MMR/MSI and p53 status.

In the case of the detection of MMRd/MSI-high, our findings are in line with those observed in other studies regarding the agreement between both methods [12,14,16,19,20]. A recent meta-analysis shows a pooled sensitivity of 96% (95% CI, 93–98%) with moderate heterogeneity among studies (I2 = 74.7%) and a pooled specificity of 95% (95% CI, 93–96%) with also minimal heterogeneity (I2 = 22.7%) for MMR IHC method. The overall accuracy, determined by area under the curve (AUC), is 99%. This meta-analysis concludes that IHC for the 4 MMR proteins is an accurate surrogate of MSI molecular testing in EC tumours [16].

To reduce the cost of the four MMR proteins test, a combination of only two antibodies, MSH6 and PMS2, is proposed with an equivalent accuracy to testing all four proteins. However, the results of this combination can lead to pitfalls in the interpretation of MMR expression due to the heterodimer character of MLH1 pairing with PMS2, and MSH2 with MSH6 [11,16,21]. The use of this combination is therefore discouraged [11], and for these reasons we did not assess its diagnostic performance.

The IHC method is recognised as the preferred method used to identify MMRd/MSI. Recognised advantages of the IHC method are: 1/the short time-frame to obtain results (1–2 days); 2/its wide availability; 3/its low cost regarding all components used in the analysis; 4/the fact that it is readily interpretable by pathologists; 5/its ability to be performed on a limited amount of tissue; 6/its correlation with morphology; 7/its feasibility for all types of formalin-fixed paraffin-embedded (FFPE) specimens (biopsy and or surgical samples); 8/its amenability to IHC external quality assurance schemes; 9/its ability to identify which MMR gene is mutated, especially in the detection of MSH6 mutations that can potentially be missed in MSI testing. Additionally, the detection of mutations in MLH1, MSH2, PMS2 and MSH6 is of major importance for screening Lynch syndrome [11,12,14,22].

However, the MMR IHC method is associated with pitfalls in its interpretation. Firstly, MMR IHC is a fixation-sensitive method. To avoid erroneous interpretation of one or more stains as loss of expression, it is important to adequately examine all fixed areas. Secondly, weak or focal MMR expression may be seen in the presence of MMR deficiency. In this case comparison with the internal control is an essential step. If the expression of MMR proteins is not strong and diffuse when compared to the internal control, the MMR expression should be noted and reported as defective or equivocal. To solve this problem a repeat staining in different sections is recommended. Thirdly, subclonal expression, defined as focal loss of expression by 10% of the tumour cells, could be observed in a minority of cases and should be assigned to the MMRd group. Also, a low proportion of MLH1-loss cases can reveal punctate nuclear expression that may be erroneously interpreted as retained/normal expression. This pattern should be reported as a loss of expression and is thought to be a technical artifact. Additionally, the MMR proteins are localised in the nucleus. In some cases, possibly related to technical reasons, there is a relatively conspicuous cytoplasmic or membranous staining in the absence of nuclear staining; such cases should be reported as abnormal. Finally, other patterns/problems may occur, such as loss of 3 or more proteins [11,21].

PCR amplification of microsatellite markers to assess MSI status (BAT-25, BAT-26, NR-21, NR-24 and NR-27) provides rapid results (1–2 days) and at low cost. In contrast, PCR requires a significant tumour cell percentage (30%) in order to perform analysis [12].

Despite a substantial agreement between MMR IHC and MSI PCR methods [23], 12% of discrepancies between both methods was observed in our study. Discrepancies can be explained by tumour heterogeneity or by an incomplete sensitivity/specificity of either method: poor DNA quality, insufficient or heterogenous antibody binding and retained expression of mutated proteins [11,14,21,24]. Indeed, analyses performed on gastrointestinal tract tumours indicate a sensitivity of around 90% for each method, and these numbers are lower for EC [12,25].

Aware of these weaknesses, some experts recommend a combination of the use of IHC and a molecular MSI method to achieve maximal sensitive and specific detection of MMRd/MSI-high tumours [12]. Indeed, as either method shows a sensitivity around 90%, the use of a single approach might miss 10% of Lynch syndromes.

Due to the risk of misclassified Lynch syndrome cases, the molecular testing used alone for MSI in EC patients is currently insufficient [12]. Even though the NGS method shows promising results in detecting MSI tumours in colorectal cancer, further studies are needed to recommend this method in other tumours of the Lynch syndrome spectrum such as EC [26].

The present study was also conducted to assess the agreement between IHC versus NGS method for the determination of the p53 status amongst EC patients. As described above, the IHC method is quick, easy to perform, and less expensive [9,13,27] when compared to the NGS method [9,13]. Our results show a moderate agreement [23] between p53 IHC and *TP53* NGS when analyses are performed after exclusion of *POLE*mut and MMRd cases, as per the WHO algorithm, with an accuracy of 81.3%. This accuracy is lower than those observed in the studies of Singh et al. and Vermij et al., who respectively noted an accuracy of 95.1% and 94.5% [9,13]. Biense et al. observed discrepancies between the two methods. In their study the risk of misclassification is in the order of 5% if the p53 status is determined only by IHC rather than NGS [17]. A recent meta-analysis shows that “overexpression or complete absence” of p53 are highly accurate immunohistochemical surrogates of *TP53* mutation detected by NGS with an AUC of 0.97. The pooled sensitivity is 83% (95% CI, 71–91%) with high heterogeneity among studies (I2 = 76.9%) and the pooled specificity is 94% (95% CI, 89–97%) with minimal heterogeneity (I2 = 4.4%) [15].

In order to achieve high diagnostic accuracy in predicting the presence of *TP53* mutation with IHC, it is important to have optimal internal and external controls to correctly interpret the p53 staining [11]. Different p53 patterns are observed in EC tumours and may be divided into “normal” or “wildtype” pattern and the “mutation-type”, “mutant”, “aberrant” or “abnormal” pattern. Abnormal patterns include overexpression of p53 in the nucleus, null or complete absence of p53 expression, cytoplasmic and subclonal p53 expression [9,28]. As shown in the results of our study, overexpression is most commonly associated with non-synonymous missense mutation in *TP53*. This mutation results in a nuclear accumulation of a degradation-resistant protein. The complete absence of p53 in tumour cells is the consequence of stop gain and splice site mutations. Lastly, the accumulation of p53 in the cytoplasm of tumour cells, without nuclear overexpression, is related to C-terminal mutations [13,29]. Nevertheless, discordant cases can be observed between p53 expression and *TP53* mutation. Among possible explanations of these discordant cases, the subclonal pattern which has recently been described, might not be detected by the NGS method because it depends on the area of DNA extraction [9]. It can explain few discordant cases, but this was not the case in our study. An overexpression of p53 protein, without underlying *TP53* mutations, can also explain discrepancies between IHC and NGS. This overexpression might be due to the dysregulation of factors such as estrogen receptor (ER) isoform ERβ and MDM2 (mouse double minute 2) [30,31].

Some other factors can lead to misclassified p53 patterns such as: 1/the cellular state of differentiation and proliferation activity which can show a wide range of staining (weak to strong staining) in wildtype pattern, 2/preanalytical factors (fixation problem, antigen degradation) or splicing mutations which can explain “mosaic” pattern, and 3/technical artifacts which result in nonspecific nuclear blush or nonspecific cytoplasmic blush. Nuclear blush could be misinterpreted as wildtype in null pattern and the cytoplasmic blush could be interpreted as p53 abnormal when it should be ignored [11,28,32]. The interpretation of p53 is also affected by the simultaneous presence of two or three molecular signatures which give heterogenous staining [11,32]. About 3% of EC cases, called “multiple classifier”, can be classed as MMRd and p53 abnormal, *POLE*mut and p53 abnormal, *POLE*mut and MMRd and p53 abnormal. In these cases the driver molecular subtype is determined as follows: *POLE*mut prevails over the MMRd and p53 abnormal signature, and MMRd prevails over the p53 abnormal signature [11,33].

To reduce the resources involved in molecular classification, in particular *POLE* NGS testing, Betella et al., propose a novel algorithm. This new algorithm consists of analysing MMR proteins and p53 using IHC method in early-stage (stage I–III) EC, which do not require *POLE* mutation analysis by NGS. In their study this new algorithm reduced the number of *POLE* sequencing tests by 67% and that of p53 IHC by 27% compared to the molecular classification of ESGO/ESTRO/ESP 2020 for EC [34]. Likewise, the British Association of Gynaecological Pathologists provides an algorithm to limit *POLE* testing to those cases where it is essential for patient care [35].

Recently, Jamieson et al. proposed a one-step DNA-based molecular classifier, ProMisE-2, to assess mutations in *POLE*, *TP53* and presence of MSI. The first results show an excellent agreement (Cohen’s kappa: 0.93) with the initial ProMisE algorithm, which uses IHC for testing MMR and p53 proteins, and its conserved prognostic value. This one-step test could be performed on pre-operative biopsy with the combined advantages of having the molecular information available at the time of EC diagnosis and reducing the number of steps needed to define the molecular risk group of EC patients. Further investigations are needed before implementation in clinical practice [36].

Furthermore, artificial intelligence is a promising solution to characterise histomorphological EC molecular subtypes and their disease prognosis [37,38,39]. Studies are ongoing in this specific field.

Our study presents some limitations and strengths. The main limitation is related to its retrospective design and the limited sample size. Additionally, the work had to be carried out with missing values, which is unavoidable in clinical research. Among strengths, pathology analyses were centralised. Furthermore, therapeutic management was constant throughout the inclusion period and followed international guidelines.

In conclusion, for the determination of MMR/MSI status, IHC and PCR showed equivalent diagnostic performance. Nevertheless, these methods give complementary information for effective management of EC. Therefore, as both methods are currently available in most cancer centres at a cost that is reasonable considering the total cost of EC care, we would recommend using both methods. Concerning the determination of p53 status, the moderate agreement observed between IHC and NGS methods requires further prospective studies to explore the prognostic and predictive values of each method and how they would affect the associated algorithm, and in turn to eventually choose one method in preference to the other. These future results could help healthcare providers and researchers to adopt a more efficient evidence-based practice.

## 4. Materials and Methods

### 4.1. Study Population and Data Collection

A retrospective cohort of EC patients who were treated in the Gynecological Department of the University Hospital of Liège in Belgium between January 2019 and December 2021 were analysed according to international guidelines. Eligibility criteria included all histological subtypes (endometrioid and non-endometrioid), all tumour grades, and all stages according to FIGO 2009. Patients with other concomitant cancer were excluded.

Clinicopathological data included age, Body Mass Index (BMI), histological subtypes, tumour grades, nodal staging and FIGO stage.

### 4.2. Immunohistochemistry and Molecular Analyses

Immunohistochemistry was performed on 4 µm thick formalin-fixed paraffin-embedded (FFPE) samples mounted on positively charged glass slides, by using the VENTANA P53 (ROCHE-CLONE DO-7), MLH1 (ROCHE-CLONE M1), MSH2 (ROCHE-CLONE G219-1129), MSH6 (ROCHE-CLONE SP93) and PMS2 (ROCHE-CLONE A16-4) antibodies on an automated BenchMark instrument (Ultra, Ventana Medical Systems, Tucson, AZ, USA). IHC expression of p53 was reported as either “normal” or “abnormal”. Normal p53 expression was defined as nuclear staining of variable intensity in 1–80% of the tumour. The p53 expression was considered “abnormal” in the following four cases: when strong nuclear staining was observed in more than 80% of the tumour (nuclear overexpression), when nuclear staining was totally absent (complete absence or null mutant), when cytoplasmic staining, without nuclear overexpression, was noticed (cytoplasmic overexpression) or when a combination of more than one pattern of staining, with each present in at least 5% of tumour cells, was observed (subclonal). [9,13,27]. An internal positive control was used in order to determine these patterns. Figure 1 shows examples of normal and abnormal p53 expression using IHC.

MMRd was defined when one or more of the four MMR proteins (MLH1, MSH2, MSH6 or PMS2) were unexpressed in the presence of an internal positive control (healthy stromal cells). If all four proteins were present, MMR was considered as “stable” or “proficient” (MMRp).

MSI status was determined using the pentaplex PCR assay described by Suraweera et al. 2002 and Buhard et al., 2004 [40,41]. Briefly, fluorescent multiplex PCR was performed for five quasimonomorphic mononucleotide repeats (NR-27, NR-21, NR-24, BAT-25 et BAT-26). One primer in each pair was labelled with one of the fluorescent markers (FAM for BAT-26 and NR-21, HEX for BAT-25 and NR-27 and TET for NR-24). All PCR conditions and primer sequences are available upon request. PCR products labelled with fluorescent dyes were analysed by an ABI 3500XL Genetic Analyzer (Applied Biosystems by Thermo Fisher Scientific, Waltham, MA, USA). Tumours were classified as MSI-High when at least 3 out of 5 mononucleotide repeats showed instability, MSI-low when one or two mononucleotide repeats showed instability, and MSI-Stable (MSS) when no mononucleotide repeats showed instability. Since MSI-low tumours should be considered as being MSS tumours [11,14], these MSI assay results have been grouped. Therefore, two groups were distinguished: MSI-High and MSS.

*POLE* and *TP53* mutation were determined by NGS. Regions of interest were amplified by multiplex PCR using Qiagen multiplex PCR plus (Qiagen, Hilden, Germany). The regions of interest included exonuclease domain (exons 3 to 14) of *POLE* (NM_006231.2) and coding regions (exons 2 to 11) of *TP53* (NM_000546.4). All PCR conditions and primer sequences are available upon request. Molecular barcoding was performed with the MID kit for Illumina Miseq (Agilent-Multiplicom, Niel, Belgium) according to the manufacturer’s recommendations. PCR products from each patient were purified using Agencourt AMPure XP beads (Beckman Coulter, Brea, CA, USA) and then quantified by qPCR using KAPA Universal Library Quantification Kit (Roche, Basel, Switzerland) and the CFX Connect Reader (Biorad, Hercules, CA, USA). These individually tagged amplicon libraries were pooled in equimolar amounts to obtain the final library. This latter was then sequenced on the Illumina MiSeq sequencing platform using a MiSeq v2 cartridge (500 cycles). Data were finally analysed using the SeqNext module (version 4.1.1) (JSI Medical systems, Ettenheim, Germany).

### 4.3. Statistical Analysis

Results were expressed as the median and interquartile range (IQR: P25–P75) for quantitative variables and as number (%) for categorical findings. The diagnostic capacity of the IHC method for the determination of MMR and p53 status according to the molecular techniques was assessed in terms of sensitivity, specificity, accuracy, PPV and NPV. All diagnostic characteristics were associated with their 95% confidence interval (95%CI). The McNemar test was used to compare paired proportions and Cohen’s Kappa coefficient with 95% CI to evaluate the agreement between IHC methods and molecular techniques for the status of both indicators.

For the first part of this study the molecular algorithm was not applied. The maximum amount of available molecular data was used to determine the diagnostic performance of IHC versus molecular techniques. For the second part the WHO algorithm was applied. Thus, if tumours presented the *POLE* mutation, whatever the MMR/MSI and the p53 status, they were allocated to the *POLE*mut subgroup. Tumours with a *POLE* wildtype and displaying a MMRd, whatever the p53 status, were classified in the MMRd subgroup. Finally, the two last subgroups were determined following the p53 status; either p53 abnormal or NSMP subgroups [5,11,33]. If one molecular feature could not be determined, and thus the molecular subgroup could not be defined, the case was excluded from the study.

Statistical calculations were always made on the maximum number of data available. Missing values were neither replaced nor imputed. Results were considered significant at the 5% critical level (*p* < 0.05). Statistical analyses were performed using SAS version 9.4 (SAS Institute, Cary, NC, USA).

## Figures and Tables

**Figure 1 ijms-24-04866-f001:**
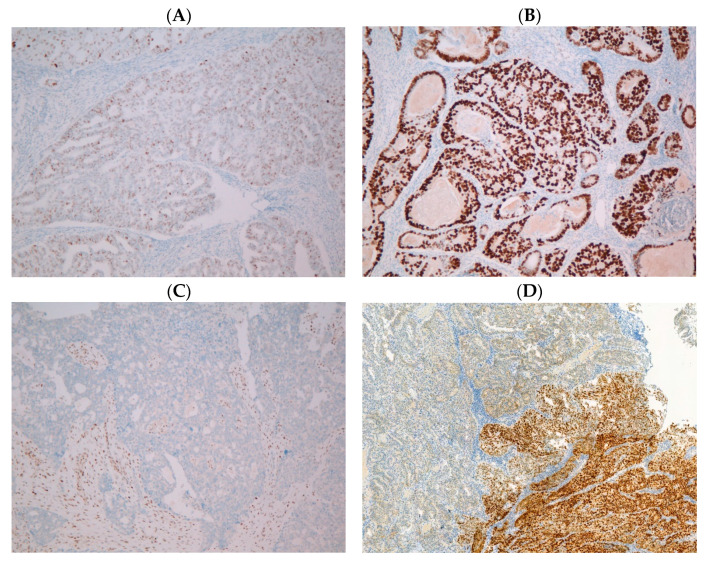
Examples of normal (**A**) and abnormal p53 expression by IHC: nuclear overexpression (**B**), complete absence (**C**) and subclonal patterns (**D**).

**Table 1 ijms-24-04866-t001:** Clinicopathological characteristics of EC patients (*n* = 132).

Variables	
Age at diagnosis, median (P25–P75) (years)	69.0 (63.0–76.0)
BMI, median (P25–P75) (kg/m^2^)	28.3 (24.5–33.3)
Histological subtypes, n (%)	
Endometrioid	107 (82.3)
Non-endometrioid	23 (17.7)
FIGO tumour grade, n (%)	
G1–G2	85 (65.9)
G3	46 (35.1)
Nodal stage, n (%)	
N0	97 (91.5)
N1–N2	9 (8.5)
FIGO stage, n (%)	
IA	48 (36.9)
IB	58 (44.6)
II-IVB	24 (18.5)

BMI, Body Mass Index; (P25–P75), interquartile range.

**Table 2 ijms-24-04866-t002:** Molecular profile of EC tumours (*n* = 132).

Variables	No. of Patients	Frequency (%)
*POLE*	113	
Wildtype		102 (90.3)
Ultra-mutated		11 (9.7)
MMR IHC	131	
MMRd		44 (33.6)
MMRp		87 (66.4)
MSI PCR	99	
MSS		65 (65.7)
MSI-high		34 (34.3)
p53 IHC, n (%)	129	
Normal		95 (73.6)
Abnormal		34 (26.4)
*TP53*, n (%)	106	
Normal		81 (76.4)
Mutated		25 (23.6)

**Table 3 ijms-24-04866-t003:** Agreement between MMR IHC status and MSI PCR testing.

		MSI PCR			
		All EC (*n* = 99)		*POLE*wt EC (*n* = 83)	
		MSI-high	MSS	MSI-high	MSS
MMR IHC status	MMRd	31	8	25	7
	MMRp	3	57	3	48
	Sensitivity %	91.2 (95% CI: 76.3–98.1) *		89.3 (95% CI: 71.8–97.7) *	
	Specificity %	87.7 (95% CI: 77.2–94.5) *		87.3 (95% CI: 75.5–94.7) *	
	Accuracy %	88.9 (95% CI: 81.0–94.3) *		88.0 (95% CI: 79.0–94.1) *	
	PPV %	79.5 (95% CI: 63.5–90.7) *		78.1 (95% CI: 60.0–90.7) *	
	NPV %	95.0 (95% CI: 86.1–99.0) *		94.1 (95% CI: 83.8–98.8) *	

* Exact Cis.

**Table 4 ijms-24-04866-t004:** The protein profile of *POLE* wildtype EC patients according to MMR IHC status and MSI PCR testing (*n* = 83).

MMR IHC	MLH1	PMS2	MSH6	MSH2	MSI PCR	Count
MMRd	1	1	0	1	MSI-high	2
MMRd	1	1	0	0	MSI-high	1
MMRd	0	0	1	1	MSI-high	21
MMRd	1	0	1	1	MSI-high	1
**MMRp**	**1**	**1**	**1**	**1**	**MSI-high**	**3**
**MMRd**	**1**	**1**	**0**	**1**	**MSS**	**1**
**MMRd**	**0**	**0**	**1**	**1**	**MSS**	**6**
MMRp	1	1	1	1	MSS	48

1 = Protein present, 0 = Protein lost.

**Table 5 ijms-24-04866-t005:** Agreement between p53 IHC and *TP53* NGS.

		*TP53* NGS			
		All EC (*n* = 104)		*POLE*wt and MMRp EC (*n* = 48)	
		Mutated	Normal	Mutated	Normal
p53 IHC	Abnormal	18	12	12	8
	Normal	6	68	1	27
Sensitivity %		75.0 (95% CI: 53.3–90.2) *		92.3 (95% CI: 64.0–1.00) *	
Specificity %		85.0 (95% CI: 75.3–92.0) *		77.1 (95% CI: 59.9–89.6) *	
Accuracy %		82.7 (95% CI: 74.0–89.4) *		81.3 (95% CI: 67.4–91.1) *	
PPV %		60.0 (95% CI: 41.6–77.3) *		60.0 (95% CI: 36.1–80.9) *	
NPV %		92.0 (95% CI: 83.2–97.0) *		96.4 (95% CI: 81.7–99.9) *	

* Exact Cis.

**Table 6 ijms-24-04866-t006:** Comparison between p53 IHC pattern and *TP53* mutation status (*n* = 48).

p53 IHC Pattern	*TP53* Mutation Status	Type of *TP53* Mutation	Counts
Abnormal-CA	Normal	-	4
Abnormal-OE	Normal	-	4
Abnormal-OE	Mutated–Exon 5	Non-synonymous missense mutation	2
Abnormal-OE	Mutated–Exon 7	Non-synonymous missense mutation	3
Abnormal-OE	Mutated–Exon 8	Non-synonymous missense mutation	5
Abnormal-CA	Mutated–Exon 8	Stop gain	1
Subclonal	Mutated–Exon 8	Stop gain	1
Normal	Mutated–Exon 8	Non-synonymous missense mutation	1
Normal	Normal	-	27

Abnormal-CA, Complete absence of p53 expression; Abnormal-OE, Nuclear overexpression of p53.

## Data Availability

The data presented in this study are available upon request from the corresponding author.

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
