# Peer review of "Diagnostic Performance of Immunohistochemistry Compared to Molecular Techniques for Microsatellite Instability and p53 Mutation Detection in Endometrial Cancer"

_ijms, 2023, doi:10.3390/ijms24054866_

Round 1
Reviewer 1 Report
The manuscript "Diagnostic Performance of Immunohistochemistry Compared to Molecular Techniques for Microsatellite Instability and p53 Mutation Detection in Endometrial Cancer" by Streel et al. is an interesting article that describes how to estimate the risk of recurrence and death for patients with endometrial cancer (EC) and how this may influence treatment decisions. The manuscript is well written but there some questions which need to answer. These are the following question: -
1. Author do not provide any images related to IHC of p53 and other respective proteins. In material and method IHC section, grading of p53 as normal and abnormal expression. When authors would show different images of the abnormal expression of p53 will make the manuscript easier to follow.
2. There is no information in the M & M section of IHC about positive and negative control uses.
3. Did the authors consider survival analysis in relation to p53 status and MSI/MMR status? This will help to determine whether the survival analysis correlates with p53 status, MSI/MMR status, or both.
4. What is the rationale of showing the profile for MLH1, PMS2, MSH6, MSH2, of the ten discordant cases is presented in table 4.
5. What does (P25-P75) signifies in table 1?
Reviewer 2 Report
This paper from one of the most involved European team in gynaecological oncology is interesting even if not new.
However, I have few remarks that I would like to see appear :
a) The methodology should be clearly improved especially the way/quantification for p53 immunostaining (WT versus MT). What is the cutt-off (percentage of positive cells, staining intensity, nuclear and/or cytoplasmic localization,... ?) and the correlation of the different patterns should be specified in details and in correlation with molecular biology.
b) Does p53 negative correlation immuno/NGS ("false negative") correspond to particular exon's mutation ?
c) Concerning MSI and the immuno/molecular profile, I have a basic problem because in most laboratories (as in this paper) molecular biology is done post immunolabeling and not in a systematic wat. Therefore, it's likely that positive and negative predictive values given by the authors reflect only a part of the truth. These data should be further discussed in detail.
Round 2
Reviewer 1 Report
The authors responded satisfactorily to the comments. The revised manuscript is now ready for publication in the prestigious MDPI-IJMS journal.